# DocPrompting: GENERATING CODE BY RETRIEVING THE DOCS

**Shuyan Zhou**[†], **Uri Alon**[†]
**Frank F. Xu**[†], **Zhiruo Wang**[†], **Zhengbao Jiang**[†], **Graham Neubig**[†‡]
[†]Language Technologies Institute, Carnegie Mellon University,
[‡]Inspired Cognition
`{shuyanzh,ualon,fangzhex,zhiruow,zhengbaj,gneubig}@cs.cmu.edu`

## ABSTRACT

Publicly available source-code libraries are continuously growing and changing. This makes it impossible for models of code to keep current with all available APIs by simply training these models on existing code repositories. Thus, existing models *inherently cannot generalize* to using unseen functions and libraries, because these would never appear in their training data. In contrast, when human programmers use functions and libraries for the first time, they frequently refer to textual resources such as code manuals and documentation, to explore and understand the available functionality. Inspired by this observation, we introduce DocPrompting: a natural-language-to-code generation approach that explicitly leverages code documentation by (1) retrieving the relevant documentation pieces given a natural language (NL) intent, and (2) generating code based on the NL intent and the retrieved documentation. DocPrompting is general: it can be applied to any programming language, and is agnostic to the underlying neural model. We demonstrate that DocPrompting consistently improves NL-to-code models: DocPrompting improves strong base models such as CodeT5 by 2.85% in pass@1 (52% relative gain) and 4.39% in pass@10 (30% relative gain) in execution-based evaluation on the popular Python `CoNaLa` benchmark; on a new Bash dataset `tldr`, DocPrompting improves CodeT5 and GPT-Neo-1.3B by up to absolute 6.9% exact match. [1]

## 1 INTRODUCTION

We address the task of natural language to code generation (NL→code): generating a code snippet, written in a general-purpose programming language such as Python or Bash, given a natural language intent. This task has seen sharply growing popularity recently due to the emergence of large language models trained on vast amounts of natural language and code (Chen et al., 2021; Xu et al., 2022; Fried et al., 2022). NL→code models facilitate programming for both professional and inexperienced programmers, by allowing programmers to write code by only expressing their higher-level intent.

Many existing code generation models either learn directly from input-output pairs provided as training data (Allamanis et al., 2015; Yin and Neubig, 2017; Iyer et al., 2018; Brockschmidt et al., 2019; Xu et al., 2020; Alon et al., 2020; Wang et al., 2021), or learn the mapping between input and output implicitly from naturally occurring corpora of intertwined natural language and code (Austin et al., 2021; Nijkamp et al., 2022). Nevertheless, all these works assume that *all libraries and function calls were seen in the training data*; and that at test time, the trained model will need to generate only *seen* libraries and function calls. However, new functions and libraries are introduced all the time, and even a seen function call can have unseen arguments. Thus, these existing models *inherently cannot* generalize to generate such unseen usages.

In contrast to these existing models, human programmers frequently refer to manuals and documentation when writing code (Nykaza et al., 2002; Lethbridge et al., 2003). This allows humans to easily use functions and libraries they have never seen nor used before. Inspired by this ability,

---

[1]Data and code are available at `https://github.com/shuyanzhou/docprompting`.

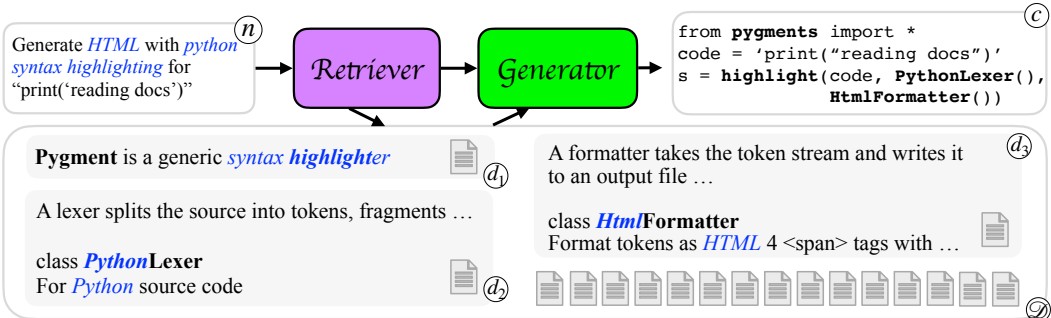

Figure 1: DocPrompting: given an NL intent $n$, the retriever retrieves a set of relevant documentation $\{d_1, d_2, d_3\}$ from a documentation pool $\mathcal{D}$. Then, the generator generates the code $c$ based on the NL and retrieved docs. DocPrompting allows the model to generalize to previously unseen usages by reading those docs. *Italic blue* highlights the shared tokens between NL and docs; **Bold** shows shared tokens between docs and the code snippet.

we propose DocPrompting: a code generation approach that learns to retrieve code documentation before generating the code. An overview of our approach is illustrated in Figure 1: First, a document *retriever* uses the NL intent $n$ to retrieve relevant code documentation $\{d_1, d_2, d_3\}$ from a documentation pool $\mathcal{D}$. Then, a code *generator* uses these docs in its prompt to generate the corresponding code $c$. The documentation pool serves as an external data store that can be updated frequently with new contents (e.g., documentation of newly released libraries), without re-training any model component. This way, DocPrompting can leverage newly added documentation, and it can generate code containing unseen and unused functions and libraries. DocPrompting is general and applicable to any programming language and underlying base architecture. To the best of our knowledge, this is the *first* demonstration of leveraging documentation in models of code explicitly and effectively.

We demonstrate the effectiveness of DocPrompting on two NL→code benchmarks and tasks, across two programming languages, and using several base models: GPT-Neo (Black et al., 2021), T5 (Raffel et al., 2020), CodeT5 (Wang et al., 2021), Fusion-in-Decoder (Izacard and Grave, 2021)), and Codex (Chen et al., 2021). Further, we experiment with both sparse retrievers such as BM25 (Robertson and Jones, 1976) and dense retrieval models such as SimCSE (Gao et al., 2021). Finally, we introduce *two new benchmarks* for retrieval-based code generation: (a) in Bash, we curate a new benchmark by crawling the `tldr` repository, and constructing the training/development/test splits without overlapping commands; (b) in Python, we re-split the popular `CoNaLa` benchmark (Yin et al., 2018) by making every test example contain at least one Python function that is not seen in the training data. Models that use DocPrompting consistently outperform their base models that generate code solely based on the NL intents. Using DocPrompting improves strong base models such as CodeT5 by 2.85% in pass@1 (52% relative gain) and 4.39% in pass@10 (30% relative gain) in execution-based evaluation in `CoNaLa`; on the new `tldr` dataset, DocPrompting improves CodeT5 and GPT-Neo-1.3B by up to absolute 6.9% exact match. We release our new benchmarks, including annotation of oracle documents for each example and pools of documentation, to serve as a test-bed for future retrieval-based code generation models.

## 2 CODE GENERATION BY READING THE DOCS

Our underlying assumption is that code documentation is the most exhaustive yet succinct resource for most libraries and programming languages (Roehm et al., 2012), and that documentation allows to effectively generalize to unseen libraries and functions (Forward and Lethbridge, 2002). We follow the retrieve-then-generate paradigm (Lewis et al., 2020; Guu et al., 2020), focusing on retrieving *documentation*. In this section, we describe the general approach of DocPrompting; in §3 and §6.2, we elaborate and experiment with practical implementations of DocPrompting.

**Formulation** Given NL intent $n$, our goal is to generate a corresponding code snippet $c$ written in some programming language (PL) such as Python. We assume that a model has access to a collection of code documentation $\mathcal{D}$. Each document $d_i \in \mathcal{D}$ describes the usage of a library, a function, or an

argument in that PL. The construction of $\mathcal{D}$ is flexible: it can either be a comprehensive set of all available libraries and functions in a PL, or a customized subset for the scope of a specific project.

## 2.1 BACKGROUND: RETRIEVAL-CONDITIONED GENERATION

Although a model may use the entire collection of documents $\mathcal{D}$, only a few documents in $\mathcal{D}$ are relevant for any particular intent. Further, it is usually computationally infeasible to directly condition on the entire, unbounded, collection of documents while making predictions. Thus, we first let the model *select* a subset of documents $\mathcal{D}_n = \{d_1, d_2, .., d_k\} \subseteq \mathcal{D}$ that are potentially relevant given $n$, and refer to this subset while generating $c$.

Overall, we decompose the probability of generating $c$ into the probability of choosing a particular subset of documents $P(\mathcal{D}_n \mid \mathcal{D}, n)$, and the probability of generating the code conditioned on the intent and the selected documents $P(c \mid \mathcal{D}_n, n)$; finally, we marginalizing over all $\mathcal{D}_n \subseteq \mathcal{D}$:

$$P(c \mid \mathcal{D}, n) = \sum_{\mathcal{D}_n \subseteq \mathcal{D}} P(c \mid \mathcal{D}_n, n) \cdot P(\mathcal{D}_n \mid \mathcal{D}, n) \tag{1}$$

assuming that $c$ is independent of $\mathcal{D}$ given $\mathcal{D}_n$ (that is, $(c \perp\!\!\!\perp \mathcal{D} \mid \mathcal{D}_n)$). Since enumerating all possible subsets $\mathcal{D}_n$ is computationally infeasible, we follow the common practice and approximate the marginalization over $\mathcal{D}_n$ in Equation (1) by taking the most probable subset of retrieved documents $\hat{\mathcal{D}}_n$, and then conditioning the prediction of $c$ on these most likely documents:

$$\hat{\mathcal{D}}_n \coloneqq \operatorname{argmax}_{\mathcal{D}_n \subseteq \mathcal{D}} P(\mathcal{D}_n \mid \mathcal{D}, n) \qquad\qquad P(c \mid \mathcal{D}, n) \approx P(c \mid \hat{\mathcal{D}}_n, n) \cdot P(\hat{\mathcal{D}}_n \mid \mathcal{D}, n) \tag{2}$$

## 2.2 DocPrompting: GENERATING CODE BY RETRIEVING THE DOCS

Equation 2 implies that DocPrompting relies of two main components: A *retriever* $\mathcal{R}$ retrieves relevant documents $\hat{\mathcal{D}}_n$ given the intent $n$; and a *generator* $\mathcal{G}$ generates the code snippet $c$ conditioned on the retrieved documents $\hat{\mathcal{D}}_n$ and the intent $n$, which compose a new prompt. Specifically, $\mathcal{R}$ computes a similarity score $s(d_i, n)$ between a intent $n$ and every document $d_i \in \mathcal{D}$. Thus, the subset $\hat{\mathcal{D}}_n \subseteq \mathcal{D}$ is the top-$k$ documents with the highest similarity scores: $\hat{\mathcal{D}}_n = top\text{-}k_{d_i \in \mathcal{D}}(s(d_i, n))$.

An overview of our approach is illustrated in Figure 1: given the intent *Generate HTML with python syntax highlighting for "print('reading docs')"*, the retriever $\mathcal{R}$ retrieves three relevant documents: $d_1$ describes the syntax highlighting library `pygments`, $d_2$ describes the class `PythonLexer`, and $d_3$ describes the `HtmlFormatter` class. Given these docs and the intent, the generator $\mathcal{G}$ generates the code snippet $c$, which uses `PythonLexer` and `HtmlFormatter` from the `pygment` library.

# 3 PRACTICAL INSTANTIATIONS OF DocPrompting

DocPrompting is a general approach that is not bound to any specific model choices, and it can be instantiated with any base retriever and generator. This section presents the concrete instantiations of $\mathcal{R}$ and $\mathcal{G}$ that we found to provide the best performance in our experiments.

## 3.1 RETRIEVER INSTANTIATION

We experiment with two main types of retrievers: *sparse* retrievers and *dense* retrievers. As our sparse retriever, we use Elasticsearch[2] with the standard BM25 (Robertson and Jones, 1976). This retriever represents documents using sparse features that rely on word frequencies, such as BM25 and TF-IDF.

As our dense retriever, we follow prior work (Chen et al., 2020; Karpukhin et al., 2020; Gao et al., 2021): given a triplet $(n, c, \mathcal{D}_n^*)$, where $\mathcal{D}_n^*$ are the oracle docs for $n$, each $d_i^+ \in \mathcal{D}_n^*$ and $n$ form a *positive* pair $(n, d_i^+)$, while each $d_j^- \notin \mathcal{D}_n^*$ and $n$ form a *negative* pair $(n_i, d_j^-)$. We train the retriever in a contrastive fashion where the similarity score of a positive pair is maximized while that of in-batch negative pairs is minimized. For a pair $(n_i, d_i^+)$, the loss function is defined as:

$$\mathcal{L}^r = -\log \frac{\exp\left(\operatorname{sim}(\boldsymbol{h}_n, \boldsymbol{h}_{d_i^+})\right)}{\exp\left(\operatorname{sim}(\boldsymbol{h}_n, \boldsymbol{h}_{d_i^+})\right) + \sum_{d_j^- \in \mathcal{B}/\mathcal{D}_n^*} \exp\left(\operatorname{sim}(\boldsymbol{h}_n, \boldsymbol{h}_{d_j^-})\right)} \tag{3}$$

---

[2]`https://github.com/elastic/elasticsearch`

where $\boldsymbol{h}_x$ is the representation of $x$ computed by a neural encoder, and $\mathcal{B}$ are positive docs for other examples in the batch. We define $\text{sim}(\boldsymbol{h}_x, \boldsymbol{h}_y)$ as the cosine similarity between $\boldsymbol{h}_x$ and $\boldsymbol{h}_y$.

We use all $(n_i, d_i^+)$ in the training set as our supervised training dataset. Additionally, we use all sentences in the documentation pool for weak supervision: Following Chen et al. (2020) and Gao et al. (2021), representations of the same sentence with different dropout masks are treated as a positive example. Instead of using either supervised or weakly supervised training as in Gao et al. (2021), we simply mix the two resulting supervision signals, and examples are randomly distributed into batches. This mixture of tasks not only facilitates the learning process (§6.2), but also reduces the engineering effort required to store and reload models for separate supervised and unsupervised training phases. We initialize the retriever encoder with either the best model of Gao et al. (2021) or the encoder of CodeT5-base (Wang et al., 2021). Additional training details are provided in Appendix C

## 3.2 GENERATOR INSTANTIATION

We experimented with a variety of generator models. We used GPT-Neo-125M, GPT-Neo-1.3B (Black et al., 2021) and Codex (Chen et al., 2021), where we concatenate the retrieved documents and the NL intent as a single, long, prompt. T5-base (Raffel et al., 2019) and CodeT5-base (Wang et al., 2021) have a shorter input size of 512 tokens, which is sometimes too short for the concatenation of multiple docs. Thus, for T5 and CodeT5 we apply the fusion-in-decoder approach (FiD; Izacard and Grave, 2021): we first concatenate the intent $n$ with each retrieved $d_i \in \hat{\mathcal{D}}_n$ and encode each $(n, d_i)$ pair independently. Then, the decoder attends to *all* encoded NL-document pairs. We finetune the generator to maximize the log-likelihood of the reference code $c$ given $n$ and $\hat{\mathcal{D}}_n$.

With Codex (Chen et al., 2021), we performed few-shot learning rather than finetuning because the model parameters are not publicly available. We constructed the prompt with three static examples, each of which is a concatenation of retrieved documentation, an NL intent and the reference code snippet. We then appended the test example and its retrieved documentation to the few-shot examples. We used the *code-davinci-001* version because we suspect potential leakage of the test set into the training set of *code-davinci-002*. See more details in Appendix H. Training details, hyper-parameter settings and example prompts can be found in Appendices E and D.

## 4 EXPERIMENTAL SETUP

We evaluate DocPrompting on two NL→code tasks: shell scripting (§4.1), in which we generate complex shell commands given an intent, and Python programming (§4.2), where we generate answers in Python for NL questions. In this section, we first introduce a *newly curated* benchmark tldr; we then describe our re-split of the popular CoNaLa benchmark (Yin et al., 2018). For each benchmark, we provide a global documentation pool $\mathcal{D}$ that is shared for all examples and oracle documents $\mathcal{D}_n^*$ which we use to train the retriever. We release our newly curated benchmarks to serve as test-bed for future retrieval-based code generation models.

## 4.1 SHELL SCRIPTING

tldr is a community-driven project that maintains easily-readable help pages with examples for over $2.5k$ Bash commands in over 25 natural languages[3]. We collected pairs of English intents and Bash command lines. The NL intents are written by human users, and the Bash commands range from popular ones like cat and tar, to uncommon commands such as toilet and faketime. Our resulting tldr benchmark contains 1,879 unique Bash commands and 9,187 NL→Bash pairs. We constructed the training, development and the test set with *completely disjoint commands* to test the generalizability of a code generation model. The shared documentation pool $\mathcal{D}$ is made up of the $400k$ paragraphs from the 1,879 Bash manuals. Each paragraph describes a single concept such as an

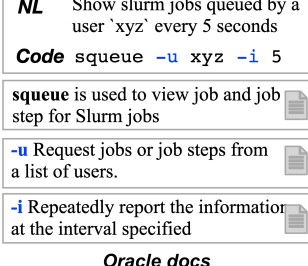

**Oracle docs**

Figure 2: An example NL-code pair from tldr, along with three oracle documentation items.

---
[3] https://github.com/tldr-pages/tldr

argument flag. We further curated the oracle documents $\mathcal{D}_n^*$ for each example using simple string matching. An example from `tldr` is shown in Figure 2. To the best of our knowledge, this is the first work to leverage `tldr` as an NL→code benchmark. Detailed statistics and additional details are provided in Appendix A. In `tldr`, each NL intent results in a single Bash command with a combination of argument flags. We therefore first retrieve an entire Bash manual; then, we take the top manual and retrieve the top-10 paragraphs from that manual.

**Evaluation metrics** We measure: (a) command name accuracy (CMD Acc) – whether the command name (*e.g.,* `cat`) is an exact match; (b) exact match (EM) – exact match between the reference and the generation; (c) token-level F1; and (d) character-level BLEU (charBLEU; Lin et al., 2018; Shi et al., 2022). In all metrics, we disregard user-specific variable names in the references and the models outputs. For example, "`mycli -u [user] -h [host] [database]`" is evaluated as "`mycli -u $1 -h $2 $3`".

## 4.2 PYTHON PROGRAMMING

`CoNaLa` (Yin et al., 2018) is a popular benchmark for NL→Python generation. NL intents are StackOverflow questions, and code snippets are their answers. Both intents and code snippets are rewritten by human annotators. We re-split the dataset to test models' generalization to unseen Python functions. In our re-split, we verifed that every example in the development or the test set uses at least one Python function (*e.g.,* `plt.plot`) that was *not* seen in the training data. In addition, we make sure that the examples from the same StackOverflow posts are in the same set to prevent leakage. This re-split results in 2,135/201/543 examples in the training/development/test sets, respectively.

The `CoNaLa` documentation pool $\mathcal{D}$ contains 35,763 documents, each describing a single function, from all Python libraries available on `DevDocs` (`https://devdocs.io`). These include built-in libraries and other popular libraries such as `numpy`. We constructed the oracle docs $\mathcal{D}_n^*$ for each example by matching all function names in the target code $c$ with docs. More details in Appendix B.

**Evaluation metrics** We follow Yin et al. (2018) and measure BLEU-4. Since we focus on generalization to unseen functions, we additionally report function name recall (*recall*) and unseen function recall (*recall_unseen*), which measures recall among function calls that do not appear in the training set. Finally, following Chen et al. (2021); Austin et al. (2021), we used the manually written unit tests from Wang et al. (2022) for 100 examples from `CoNaLa`'s test set and measure pass@$k$. We followed Chen et al. (2021) and performed nucleus sampling (Holtzman et al., 2019) with $p = 0.95$. For each $k$, we searched for the best temperature for each model from $\{0.2, 0.4, 0.6, 0.8, 1.0\}$. On average, each example has 2.03 tests. The concatenation of multiple Python docs often exceeded the length limit of GPT-Neo, we hence experimented in this dataset with FiD, which allows longer inputs. Additional details are provided in Appendix B.

## 5 RESULTS

In all following results, all models with DocPrompting use the top-10 retrieved docs from the best retriever on that dataset (Table 4). Every baseline uses the exact same setup as its "+DocPrompting" version, except for not using the documentation.

### 5.1 SHELL SCRIPTING RESULTS

Results for `tldr` are shown in Table 1. DocPrompting consistently improves the base models. For example, T5+DocPrompting achieves more than *twice* higher accuracy in predicting the command name, more than 16 charBLEU points on the entire prediction, and almost 9% of absolute exact match gain, compared to the vanilla T5. In the few-shot learning setting with Codex, DocPrompting brings gains of 6.7 charBLEU points, and consistent improvement across all metrics over the baseline that observes only NL-code pairs in its prompt. These results show that retrieving documentation also benefits strong models such as Codex, and with only few examples in the context.

**Code generation with oracle command names** In realistic settings, a human programmer may know the command name they need to use (*e.g.,* `awk`), but not know the exact usage and flags. In fact, better understanding of the usage of *known* commands is the purpose of Unix `man` pages and the

Table 1: Results on shell scripting, using a BM25 retriever with top-10 retrieved docs, on the test set of `tldr`. For the "oracle command name" experiments, we selected the best model of each type.

| Model | | CMD Acc (%) | EM (%) | Token F1 | charBLEU |
|---|---|---|---|---|---|
| GPT-Neo-125M | - | 11.96 | 1.94 | 28.75 | 19.99 |
| | +DocPrompting | **25.32** | **3.56** | **31.23** | **24.43** |
| GPT-Neo-1.3B | - | 14.55 | 3.12 | 32.46 | 24.70 |
| | +DocPrompting | **27.59** | **9.05** | **37.24** | **30.57** |
| T5 | - | 10.02 | 0.76 | 19.90 | 25.48 |
| | +DocPrompting | **30.28** | **9.16** | **37.58** | **31.97** |
| CodeT5 | - | 14.60 | 2.18 | 30.00 | 21.50 |
| | +DocPrompting | **30.72** | **9.15** | **36.71** | **33.83** |
| Codex *3-shots* | - | 27.48 | 8.94 | 36.04 | 16.94 |
| | +DocPrompting | **31.21** | **9.29** | **36.77** | **23.72** |
| With the oracle command name | | | | | |
| T5 | - | - | 12.96 | 59.36 | 45.05 |
| | +DocPrompting | - | **22.55** | **64.84** | **54.28** |
| Codex *3-shots* | - | - | 22.44 | 62.26 | 50.29 |
| | +DocPrompting | - | **32.43** | **69.73** | **55.21** |

Table 2: Comparison to approaches that retrieve examples (Parvez et al., 2021; Pasupat et al., 2021).

| Model | | CMD Acc (%) | EM (%) | Token F1 | charBLEU |
|---|---|---|---|---|---|
| GPT-Neo-125M | +ExPrompting | 6.68 | 0.32 | 20.49 | 11.15 |
| | +DocPrompting | **25.32** | **3.56** | **31.23** | **24.43** |
| GPT-Neo-1.3B | +ExPrompting | 14.01 | 2.8 | 30.07 | 22.11 |
| | +DocPrompting | **27.59** | **9.05** | **37.24** | **30.57** |

`tldr` project. We conducted an oracle experiment where we provided T5 (which was the strongest model using DocPrompting) and Codex with the oracle command name (*e.g.,* `awk`). This oracle information is provided to both the baseline and the model that uses DocPrompting. The results are shown on the bottom part of Table 1. When the oracle command is given, DocPrompting further improves over the base models. For example, when providing Codex with the ground truth command name, DocPrompting improves its exact match from 22.44% to 32.43%.

**Should we retrieve documentation or examples?** All existing retrieval-based models of code retrieve NL-code pairs or code snippets, rather than documentation. To simulate this scenario, we followed Parvez et al. (2021) and Pasupat et al. (2021) to retrieve NL-code pairs from the training set of `tldr`, and refer to this baseline as `ExPrompting`. We finetuned the best retriever RoBERTa and two generators, and retrieved the top-30 NL-code pairs for every example. As shown in Table 2, *retrieving documentation* (DocPrompting) provides much higher gains than retrieving examples (`ExPrompting`). Theoretically, adding examples of unseen commands can help `ExPrompting` generalize to them as well. However, new libraries and functions may not have available examples on the web yet, while documentation often *does* becomes available when the library is released.

## 5.2 PYTHON PROGRAMMING RESULTS

Table 3 shows the results on `CoNaLa`. CodeT5+DocPrompting yields a 1.65 BLEU improvement over the state-of-the-art baseline that was initialized with CodeT5.[4] When measuring the recall of the generated function names, the benefit of DocPrompting is especially higher for *unseen* functions (*recall$_{unseen}$*). For example, DocPrompting achieves 18.30 compared to only 9.03 of the base CodeT5 in unseen functions. Additionally, DocPrompting improves in-context learning setting with Codex.

---

[4]In a separate experiment on the original split of `CoNaLa`, this baseline achieved a BLEU score of 39.12, which outperforms the previous state-of-the-art (Beau and Crabbé, 2022) by 4.92 BLEU points.

Table 3: Results on `CoNaLa`, using a CodeT5 retriever with top-10 retrieved docs. Function recall (Recall) measures how many functions in the reference code are correctly predicted, and unseen function recall ($Recall_{unseen}$) only considers the subset held out from the training data.

| Model | | BLEU | Recall | $Recall_{unseen}$ |
|---|---|---|---|---|
| Codex *3-shots* | - | 43.16 | 39.52 | - |
| | + DocPrompting | **43.47** | **39.87** | - |
| | + DocPrompting oracle docs | 50.59 | 57.84 | - |
| T5 | - | 28.07 | 14.36 | 2.57 |
| | + DocPrompting | **30.04** | **21.34** | **8.24** |
| CodeT5 | - | 34.57 | 24.24 | 9.03 |
| | + DocPrompting | **36.22** | **27.80** | **18.30** |
| | + DocPrompting oracle docs | 49.04 | 72.20 | 63.91 |

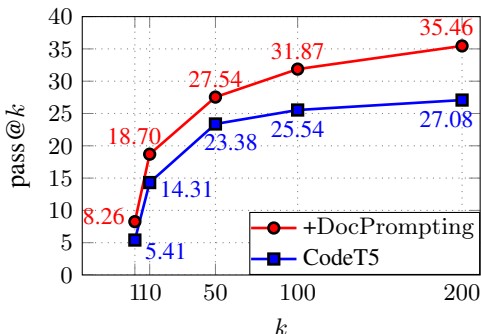

Figure 3: Pass@$k$ of CodeT5 with and without DocPrompting on 100 `CoNaLa` examples.

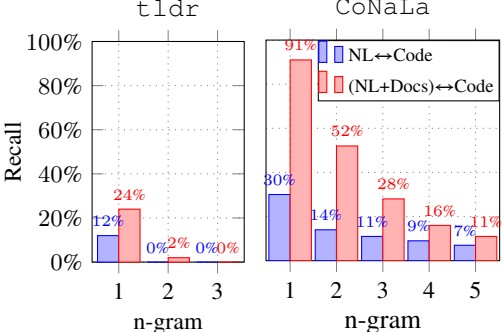

Figure 4: Using documentation significantly increases the $n$-gram overlap recall between the input and the output, in `tldr` and `CoNaLa`.

We hypothesis that the minor gain is mainly due to the potential data leakage of Codex, which violates the split of seen and unseen functions. Another reason is that a strong generator such as Codex may require an equally strong retriever as well. We find that Codex can achieve even higher results with an oracle retriever, which shows the potential further improvement by improving the retrievers. Finally, CodeT5 performs better than T5, with and without using DocPrompting. This emphasizes the importance of using code-specific pretrained models.

**Execution-based evaluation** The results are shown in Figure 3. Using DocPrompting consistently outperforms the baseline CodeT5 for all values of pass@$k$. For example, DocPrompting yields 2.85% improvement on pass@1 and 4.45% improvement on pass@5, which are realistic numbers of completions that can be suggested in an IDE. When $k = 200$, DocPrompting widens the gap to 8.38%. These results demonstrate that DocPrompting does not only improve the quality of the generated code in its surface form, but also increase its functional correctness. Additional details and results are provided in Appendix G.

## 6 ANALYSIS

### 6.1 WHY DOES READING THE DOCUMENTATION HELP GENERATING MORE ACCURATE CODE?

We believe that one of the major reasons is that *documentation eases the mapping between NL intents and code*, since the documentation contains both NL descriptions *and* function signatures. We calculated the n-gram overlap between the NL intents and their corresponding code snippets (NL↔code), and the overlap between the NL intents with their top-10 retrieved documents and their code snippets ((NL+docs)↔code). As shown in Figure 4, adding documentation *significantly increases* the overlap across $n$-grams, and increase, for example, the unigram overlap from 12% to

Table 4: Retrieval performance of multiple models on the dev set of `tldr` (top) and `CoNaLa` (bottom). RoBERTa is the best model taken from from Gao et al. (2021), and CodeT5 is the encoder of CodeT5-base (Wang et al., 2021). Models with the subscript "off-shelf" are the off-the-shelf models, and the other models were finetuned with the objective in Equation 3. The last column is the best model (RoBERTa for `tldr` and CodeT5 for `CoNaLa`) trained without the weak supervision corpus.

|  | n | BM25 | RoBERTa$_{\text{off-shelf}}$ | RoBERTa | CodeT5$_{\text{off-shelf}}$ | CodeT5 | Best w/o weak sup. |
|---|---|---|---|---|---|---|---|
| `tldr` | 1 | **32.81** | 17.53 | 30.03 | 10.45 | 18.10 | 28.30 |
|  | 5 | 51.73 | 37.89 | **52.50** | 20.26 | 38.52 | 50.50 |
|  | 10 | 59.86 | 46.80 | **60.33** | 25.73 | 51.03 | 59.84 |
|  | 20 | 62.01 | 56.11 | **64.30** | 33.65 | 57.26 | 62.30 |
| `CoNaLa` | 1 | 3.01 | 4.46 | 13.49 | 4.60 | **16.54** | 10.51 |
|  | 5 | 7.16 | 7.58 | 26.38 | 8.63 | **42.35** | 21.15 |
|  | 10 | 9.73 | 10.93 | 34.86 | 12.25 | **55.81** | 29.34 |
|  | 20 | 11.46 | 13.89 | 45.46 | 18.46 | **66.79** | 42.21 |

24% in `tldr`. That is, one of the reasons that retrieving documentation helps generating accurate code is that documentation bridges the gap between the "intent terminology" and the "code terminology".

## 6.2 ABLATION STUDY

We compared different configurations of the retriever, to gather more insights for effective DocPrompting. Table 4 shows a comparison between different retrievers and their setups. First, the performance of BM25 varies among datasets: In `tldr`, BM25 matches the recall of trained dense retrievers; however in `CoNaLa`, BM25 achieves only recall@10 of 9.73%, and strong dense retrievers such as the encoder of CodeT5 achieve recall@10 of 55.81. We hypothesize that this difference between datasets stems from the ways these datasets were created: `tldr` intents were written based on existing Bash commands and manuals; while `CoNaLa` examples were mined from StackOverflow posts, where users ask questions with limited or no context. Thus, NL intents in `CoNaLa` require a better semantic alignment with the documents, and thus benefit from dense retrievers. The gap resulting from different data curation processes was also observed by Rodriguez and Boyd-Graber (2021) in open-domain question answering (QA).

Second, retrievers that were pretrained on the target programming language are generally stronger. For example in `CoNaLa`, CodeT5 which was pretrained on Python, is both a better off-the-shelf retriever and a better finetuned-retriever than RoBERTa, which was pretrained mainly on text. In contrast, `tldr` is based on Bash, which neither CodeT5 nor RoBERTa were explicitly pretrained on. Thus, `tldr` benefits mostly from BM25 and RoBERTa rather than CodeT5 as retrievers.

Finally, training the retriever using weak supervision on the documentation pool (Section 3.1) dramatically improves the retriever. The recall of the best retrievers of each dataset without this corpus is shown in the last column of Table 4 ("Best w/o weak sup."). On `CoNaLa`, removing this corpus results in severe performance degradation. One possible explanation is that this weak supervision helps the retriever perform domain adaptation more effectively.

## 6.3 CASE STUDY

We examine the models' outputs and show two representative examples in Table 5. In the first example, `Image.open` was not seen in the training set, and the baseline CodeT5 incorrectly predicts `os.open`. In contrast, using DocPrompting allows to retrieve the docs and to correctly predict `Image.open`. In the second example, `df.to_csv` was not seen in training, and the baseline CodeT5 fails to correctly predict it. In contrast, DocPrompting *does* predict most of the `df.to_csv` call correctly, thanks to the retrieved docs. Nevertheless, DocPrompting generates an incorrect argument `skiprows=1`, instead of `header=False`. The reason is that along with the retrieved documentation of `df.to_csv`, the retriever also retrieved the documentation of `df.read_csv`, which has a `skiprows` argument. That is, the generator uses an argument of `df.read_csv` with the function `df.to_csv`. Further improving the retrievers and the generators, and post-filtering based on the validity of argument names, may mitigate such mistakes.

Table 5: Examples of predictions from `CoNaLa`, of the base CodeT5 compared to CodeT5+DocPrompting. Unseen functions are underscored.

| | |
|---|---|
| NL Intent: **Open image "picture.jpg"** | |
| Ground truth: | `img = Image.open('picture.jpg') \n Img.show` |
| CodeT5: | `os.open('picture.jpg', 'r')` |
| CodeT5+DocPrompting: | `image = Image.open('picture.jpg', 'rb')` |
| NL Intent: **Exclude column names when writing dataframe 'df' to a csv file 'filename.csv'** | |
| Ground truth: | `df.to_csv ('filename.csv', header=False)` |
| CodeT5: | `df.drop(['col1', 'col2'], axis=1, inplace=True)` |
| CodeT5+DocPrompting: | `df.to_csv('filename.csv', skiprows=1)` |

# 7 RELATED WORK

**Code generation** The most common practice in NL→code generation is training a model on a dataset of NL-code pairs (Allamanis et al., 2015; Yin and Neubig, 2017; Rabinovich et al., 2017; Iyer et al., 2018). Nevertheless, all these works assume that their training corpus covers *all* required libraries and functions, and their models are inherently incapable of generating libraries and functions that were not seen in the training data. On the contrary, DocPrompting allows models to generate calls to unseen function, by retrieving these functions' documentation and reading them at test time. Hayati et al. (2018); Parvez et al. (2021); Hashimoto et al. (2018) and Lu et al. (2017) learn to retrieve examples at test time; Pasupat et al. (2021) also considered settings where the test data has a distribution shift from the training data. However, when new libraries are released they often come with documentation, and thus we assume that documentation for new libraries is much more likely to be available than concrete natural language intent and code snippet pairs $(n, c)$ that use these libraries already. The models of Shrivastava et al. and Wu et al. (2021) retrieve code snippets from relevant files in the same project; contrarily, when predicting new libraries and functions that are *external* to the user's project, documentation is the source that is the most likely to be available.

**Retrieval augmented generation** The paradigm of retrieve-then-generate has gained popularity in the field of open-domain question answering (Guu et al., 2020; Lewis et al., 2020; Karpukhin et al., 2020), where the answer for an open-domain question exists in only few documents out of a much larger pool. Although DocPrompting takes a similar approach, documentation retrieval in code generation is even more valuable, since code libraries are updated constantly, and new libraries are introduced daily. Thus, DocPrompting allows updating the documentation pool frequently with new contents, without re-training any model components.

**Documentation conditioned generation** The model of Zhong et al. (2019) reads documents to understand environment dynamics in a grid-world game, and Branavan et al. (2011) controls situated agents in a game (Civilization II) by reading the game's manual. However, all their models were tailored to specific games; in contrast, DocPrompting is general and is applicable for a variety of programming languages and datasets.

# 8 CONCLUSION

We propose DocPrompting, a simple and effective approach for code generation by retrieving the relevant documentation. DocPrompting consistently improves NL→code models in two tasks, in two PLs, and across multiple strong base models. DocPrompting improves strong base models such as CodeT5 by 2.85% in pass@1 (52% relative gain) in execution-based evaluation on the popular Python `CoNaLa` benchmark; on a new Bash dataset `tldr`, DocPrompting improves CodeT5 and GPT-Neo-1.3B by up to 6.9% exact match, and Codex by 6.78 charBLEU score.

These results open a promising direction for NL→code generation. We believe that our results can be further improved using more clever encoding of the structured nature of long documents, and using joint training of the retriever and the generator, which hopefully will avoid cascading errors. Further, we believe that the principles and the methods presented in this paper are applicable to additional code-related tasks, and other documentation-like resources such as tutorials and blog posts. To these ends, we make all our code, data, and models publicly available.

## 9 ACKNOWLEDGEMENT

We thanks the anonymous reviewers for their useful comments and suggestions. This work is supported by a gift from Amazon AI and a contract from the Air Force Research Laboratory under agreement number FA8750-19-2-0200. The U.S. Government is authorized to reproduce and distribute reprints for Governmental purposes notwithstanding any copyright notation thereon. The views and conclusions contained herein are those of the authors and should not be interpreted as necessarily representing the official policies or endorsements, either expressed or implied, of the Air Force Research Laboratory or the U.S. Government.

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

## A    TLDR: A NEWLY CURATED SHELL SCRIPTING BENCHMARK

**NL→Bash pairs**    For each command (*e.g.,* `cat`), users contribute examples of pairs of NL descriptions and bash code (mainly one-liners), including various flags and arguments, which cover the common usages of that command. An example is shown in Figure 2.

We crawl NL-code pairs from the markdown files[5] in the `linux` and `common` folders. We discard Bash commands whose manual is unavailable (discussed below). The detailed statistics are shown in Table 6. On average, each command has 4.84 NL→Bash pairs and there is a total of 9187 NL-code pairs. To test the generalizability of a model, we construct the training, development and the test set *with completely different commands*.

Table 6: The statistics of the `tldr` shell scripting benchmark

|       | # Commands | NL→Bash pairs |
|-------|-----------|---------------|
| train | 1315      | 6414          |
| dev   | 376       | 1845          |
| test  | 188       | 928           |
| total | 1879      | 9187          |

**Documentation pool** $\mathcal{D}$    We take the bash manual of the 1897 bash commands in `tldr` to construct a documentation pool. We search each command name at `manned.org`[6], a website which archives Unix manual pages (the same as the Unix '`man <command>` command), and then extract the text contents from the returned manual page. We further break each manual into multiple paragraphs by line breaks so that each paragraph delicately describes a single concept such as a command functionality or a flag usage. We make this decision due to the large volume of content each manual has, which is too long to fit the length limitation of a neural model, and too noisy and distracts the model with irrelevant information. This results in $400k$ individual entries in the pool in total.

**Oracle manual** $\mathcal{D}_i^*$    We find the ground truth documentation for each $(n, c)$ pair through command name and flag matching heuristics. For instance, given a code snippet `toilet 'input_text' -f 'font_filename'`, we constrain our search to the documentation from `toilet` manual page and select documentation that starts with `-f` flag as an oracle paragraph. Along with the first paragraph that commonly summarizes a command, these paragraphs forms $\mathcal{D}_n^*$.

**Evaluation metrics**    We use four evaluation metrics to measure the quality of the generated code: (a) command name accuracy (*CMD Acc*) – measures whether the command name (*e.g.,* `cat`) is predicted correctly; (b) token-level F1 – converts the reference code and the generated code to bag of words and measures the token-level precision, recall, and F1 overlap; (c) exact match (EM) – measures the exact match between the reference and the generation; and (d) character-level BLEU (charBLEU; Lin et al., 2018; Shi et al., 2022). For token level F1, exact match, and charBLEU, we disregard all user-specific variables in the references and the system outputs. For example, "`mycli -u [user] -h [host] [database]`" is converted into "`mycli -u $1 -h $2 $3`". This is mainly because the variables are not instantiated in `tldr` and the style of the placeholder varies among contributors. For example, some contributors might write `[user]` as `[username]` or `[your_name]`. Therefore, measuring the surface form of user-specific variable names is less meaningful.

## B    RE-SPLITTING CONALA

**NL→Python pairs**    We adapt the popular `CoNaLa` benchmark and re-split the dataset to test the generalization scenario. This re-split makes every example in the development and the test set have at least one Python function (*e.g.,* `plt.plot`) that was not seen in the training data. There are 2135, 201, and 543 examples in the training, development and test sets, respectively. We follow the original work Yin et al. (2018) to evaluate the system outputs with BLEU-4. Since we focus on the generalization setting, we additionally report unseen function accuracy, which measures the percentage of correctly predicted held-out functions that do not appear in the training set.

---

[5] *e.g.,* `https://github.com/tldr-pages/tldr/blob/main/pages/linux/toilet.md`
[6] `https://manned.org`

**Human-annotated unit tests**    Following Chen et al. (2021) and Austin et al. (2021), we conduct execution-based evaluation on `CoNaLa` to measure the functional correctness of the generated code. We randomly selected 100 examples from the test set and manually annotated unit test for each example. For example, we wrote tests such as `assert gen_code("abcds", 2) == 4` and `assert gen_code("abde", 2) == -1` to verify whether the function `gen_code` could perform "*find the index of sub string 's' in string 'str' starting from index 2*". Each example was annotated by a single annotator. The annotation was done by two authors of the paper who program with Python daily. On average, we annotate 2.03 unit tests for each example.

**Documentation pool** $\mathcal{D}$    Our documentation pool contains 35763 manuals. These functions are from all Python libraries that are available on `DevDocs`[7]. These libraries contains the Python built-in library, and popular libraries like `numpy` and `pandas`. The documentation on `DevDocs` are curated and further transformed and indexed to allow for quick searching of APIs. We then extract each API signature and the corresponding documentation in every library, remove any content in the documentation that is not text, and segment the documentation into multiple paragraphs based on the `<p>` HTML tags. The documentation pool then contains pairs of the API signature and a single paragraph in the corresponding documentation. Although the documentation pool is not comprehensive to cover all Python libraries and functions, we find it has a high coverage rate on the `CoNaLa` dataset. This choice reflects the flexibility of our approach upon the characteristics of a target scenario.

**Oracle manual** $\mathcal{D}_i^*$    To find the oracle documents for a given NL intent $\mathcal{D}_i^*$ from the original $(n, c)$ example, we first index the function names with absolute path (*e.g.,* `plot` is indexed with `matplotlib.pyplot.plot`) with Elasticsearch. Then we query the search engine with clean version of $c$ where variable name are removed. The top-5 functions after de-duplication are treated as oracle manuals $\mathcal{D}_i^*$.

**Natural language and code associations during pretraining**    Despite our efforts, it is possible that some of the held-out functions in the test set were seen to associate with NL contexts (*e.g.,* comments) during the pretraining of a retriever and a generator. Since the generators were initialized from the same checkpoint in both the baselines and the DocPrompting models, such a possible association is expected to equally help both models. In the retriever, such a possible association did not cause the retriever to see the *exact NL intents* together with the corresponding documentation, and thus the matching between NL↔doc was not leaked. However, it is possible that there had been semantically similar intents seen along with the code snippets of the held-out functions. Nevertheless, such co-occurrence is "indirect" and "unsupervised".

## C    DENSE RETRIEVER TRAINING

We finetune the model for 10 epochs with batch size of 512 and learning rate of $1e-5$. Since CodeT5 does not use `[CLS]` token, we alternatively take the average of the hidden state of the last layer as the text representation. For `CoNaLa`, we also use the first $100k$ "mined" examples provided as part of `CoNaLa` as the supervised corpus. For `CoNaLa`, we only apply a single search step because each code snippet commonly contains more than one function. We also observed that using the first sentence that normally summarizes the usage of a function achieve the best retrieval performance than other alternatives such as using the first paragraph, or simply truncating to the maximum token length. The training takes up to 15 hours on a single A6000 GPU.

## D    GENERATOR TRAINING

We train our single-source generators for 20 epochs with learning rate $4e-5$. We train our FiD-based generators for 10000 steps. The doc length is set to 200, any further content will be truncated. We follow (Izacard and Grave, 2021) to set learning rate to $5e-5$ with 2000 steps warmup and linear learning rate decay. The batch size is set to 8. The best model is selected based on the token-level F1 score on the development set for `tldr` and BLEU score for `CoNaLa`. The training takes  8 hours on a single A6000 GPU.

## E    CODEX PROMPTS

For the baseline, we prompt Codex with three NL-code pairs and append the test query to the end. An example on `tldr` is shown on top of Table 7. On the bottom, we list the prompt with DocPrompting where documentation is provided along too. In the oracle command name setting, we prepend the command name before each NL

---

[7]`https://devdocs.io`

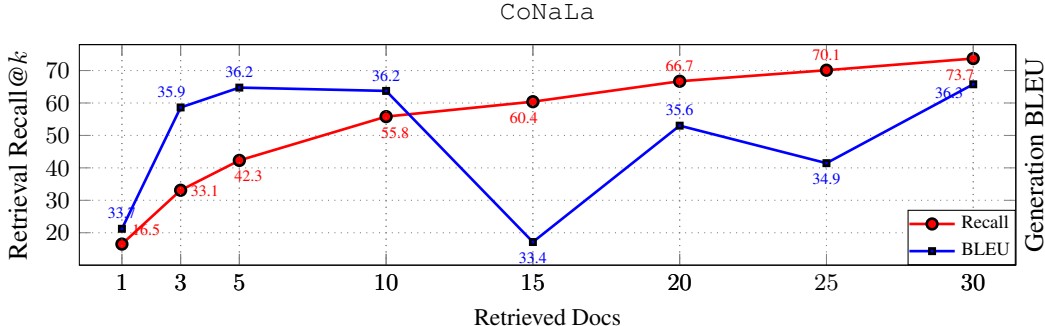

Figure 5: The recall@$k$ (%) and the corresponding BLEU score by using these top-$k$ docs on CoNaLa dataset (using CodeT5).

intent for the baseline prompt. For DocPrompting prompt, we replace the potential docs with the retrieved docs from the oracle manual.

## F ADDITIONAL ANALYSIS

**Parameter efficiency** As shown in Table 1, under a given parameter budget, we find that DocPrompting mostly benefits from parallel encoding (FiD). For example, the parallel encoding T5+DocPrompting (220M parameters) significantly outperforms the 125M parameters joint encoding Neo-125M+DocPrompting. Only scaling up Neo+DocPrompting to 1.3B parameters manages to match the 220M parameter T5+DocPrompting. A possible explanation is that although the base Neo-1.3B (without DocPrompting) generally performs better than the base T5 (without DocPrompting), parallel encoding allows to utilize the retrieved documents better, since documents are encoded independently on the encoder side.

**The impact of the number of documents** Figure 5 shows the recall@$k$ and the BLEU score compared to $k$, the number of retrieved documents. Increasing $k$ consistently yields a higher recall; however, as more irrelevant documents are retrieved, the generator cannot effectively distinguish them from the relevant ones and the overall performance remain similar. For example, CodeT5 achieves the highest BLEU score using $5 \leq k \leq 10$. In contrast, when the generator is provided with the oracle docs only, its BLEU score reaches 49.04 (Table 3). This suggests that both precision and recall of docs are important, and the benefit of using larger values of $k$ in open domain QA (Izacard and Grave, 2021) does not necessarily hold in code generation.

**Full $n$-gram overlap** Table 8 shows that using documentation significantly increases the $n$-gram overlap recall between the input and the output, in tldr and CoNaLa. Since we used BM25 to retrieve docs in tldr, the NL↔Retrieved docs overlap is high by construction. In CoNaLa, the NL↔Retrieved docs unigram overlap is high as well, but since we used a *dense* retriever, the general n-gram overlap does not have to be high for DocPrompting to work well.

**Retrieval latency** Although retrieving docs results in additional test-time computation, the increase in latency is not prohibitive. First, encoding the input for the retrieval step "costs" a *single forward pass* through the retriever's encoder, which is significantly less expensive than generation (which requires multiple time steps of the decoder). All the documentation in the retrieval pool can be encoded in advance, and finding the top-$k$ results can be performed quickly using libraries such as FAISS Johnson et al. (2019) on the GPU or ScaNN Guo et al. (2020) on CPU. The cost of this top-$k$ search is sub-linear in the size of the document pool. Second, the additional input to the generator results in an increased memory consumption, but only a small increase in latency since the tokens of a given input can be encoded in parallel. If this difference is crucial in practical settings, we can decrease the number of retrieved documents. Figure 5 shows that retrieving as few as five docs may be sufficient in many cases.

## G FULL PASS@$k$ PLOTS

In the main execution-based evaluation, pass@$k$ results in Section 5.2 and Figure 3, we took the best temperature for every model and value of $k$. Here, we show all the pass@$k$ plots with different temperatures in Figure 6.

---

# get the label of a fat32 partition
```
fatlabel /dev/sda1
```
# END

# display information without including the login, jcpu and pcpu columns
```
w --short
```
# END

# sort a csv file by column 9
```
csvsort -c 9 data.csv
```
# END

# search for a package in your current sources

---

Potential document 0: fatlabel will display or change the volume label or volume ID on the MS- DOS filesystem located on DEVICE ...

# get the label of a fat32 partition
```
fatlabel /dev/sda1
```
# END

Potential document 0: w displays information about the users currently on the machine, and their processes. The header shows, in this order ...

Potential document 1: -s, –short Use the short format. Don't print the login time, JCPU or PCPU times.

# display information without including the login, jcpu and pcpu columns
```
w --short
```
# END

Potential document 0: Sort CSV files. Like the Unix "sort" command, but for tabular data

Potential document 1: usage: csvsort [-h] [-d DELIMITER] [-t] [-q QUOTECHAR] [-u 0,1,2,3] [-b] [-p ESCAPECHAR] ...

Potential document 2: optional arguments: -h, –hel show this help message and exit -n, –names Display column names and indices from the input CSV and exit. -c COLUMNS ...

Potential document 3: csvsort -c 9 examples/realdata/FY09_EDU_Recipients_by_State.csv

Potential document 4: csvcut -c 1,9 examples/realdata/FY09_EDU_Recipients_by_State.csv — csvsort -r -c 2 — head -n 5

# sort a csv file by column 9
```
csvsort -c 9 data.csv
```
# END

Potential document 1: ...

Potential document 2: ...

...

# search for a package in your current sources

---

Table 7: Top: baseline Codex prompt with three NL-code pairs and a test intent. Bottom: DocPrompting prompt for Codex. In each in-context learning example, the oracle docs, the NL intent and the corresponding bash command are provided. We use up to five oracle docs for these examples. For a test example, the top-5 paragraphs from the retriever are represented with the NL intent. The documents' contents were omitted ("...") to save space.

Table 8: $n$-gram overlap between different contents (%). Using documentation significantly increases the $n$-gram overlap recall between the input and the output, in `tldr` and `CoNaLa`.

| `tldr` | 1 | 2 | 3 | | `CoNaLa` | 1 | 2 | 3 | 4 | 5 |
|---|---|---|---|---|---|---|---|---|---|---|
| NL↔Code | 12 | 0 | 0 | | NL↔Code | 30 | 14 | 11 | 9 | 7 |
| (NL+retrieved docs)↔Code | 24 | 2 | 0 | | (NL+retrieved docs)↔Code | 91 | 52 | 28 | 16 | 11 |
| NL↔Retrieved docs | 39 | 8 | 3 | | NL↔Retrieved docs | 72 | 14 | 3 | 1 | 1 |

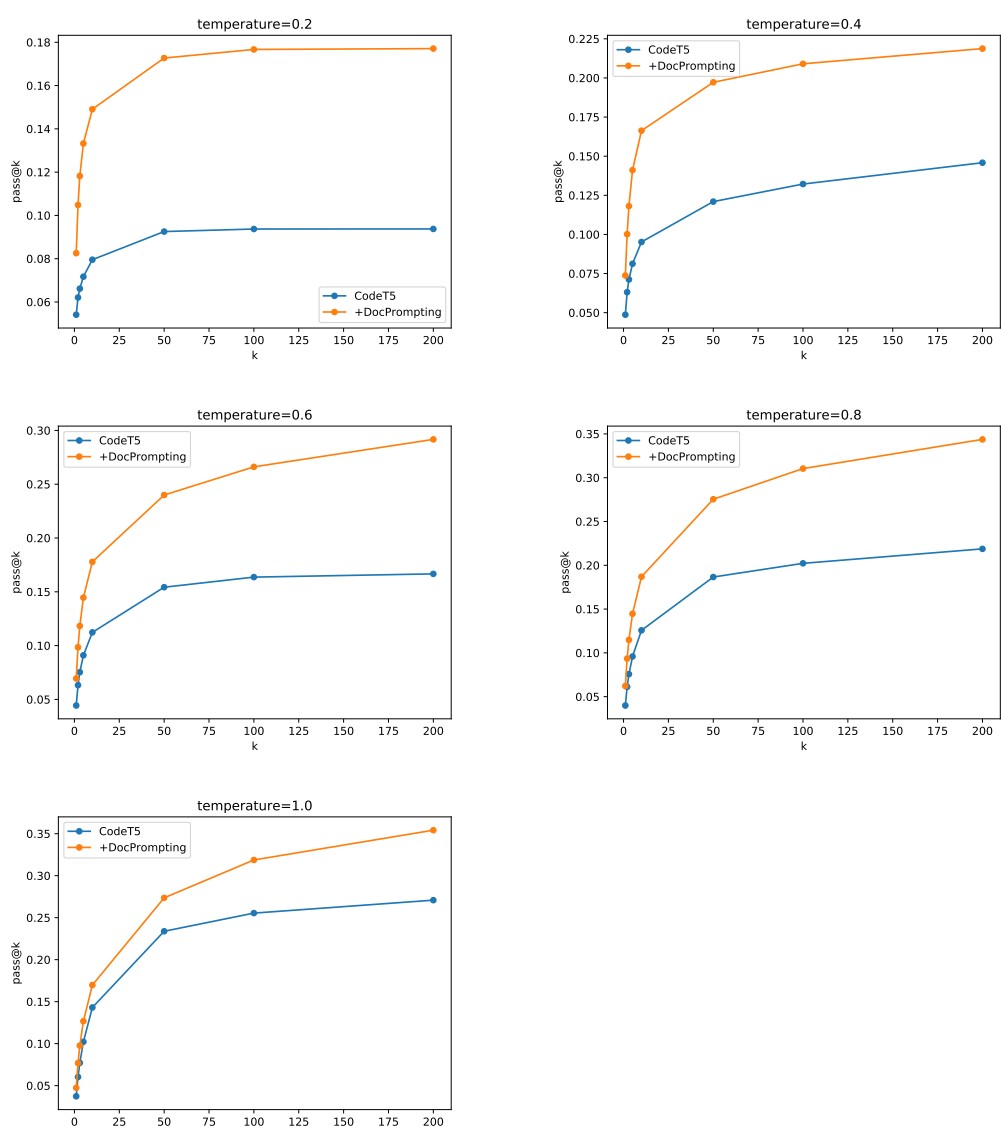

Figure 6: Pass@$k$ on 100 examples on the test set with different temperatures.

Table 9: Results on `tldr` and `CoNaLa` with *code-davinci-002*.

| Model | | tldr | | | |
|---|---|---|---|---|---|
| | | CMD Acc (%) | EM (%) | Token F1 | charBLEU |
| Codex | - | **39.01** | **14.55** | **44.89** | **33.93** |
| *3-shots* | +DocPrompting | 36.10 | 13.97 | 42.55 | 32.93 |
| With the oracle command name | | | | | |
| | - | - | 20.22 | 59.22 | 38.14 |
| | +DocPrompting | - | **33.15** | **68.59** | **44.76** |

| | CoNaLa | |
|---|---|---|
| | BLEU | Recall |
| - | **48.39** | 43.35 |
| + DocPrompting | 47.21 | **44.70** |
| + DocPrompting oracle docs | 54.67 | 59.68 |

## H    EXPERIMENTS WITH *code-davinci-002*

The results with *code-davinci-002* under few-shot learning setting is shown in Table 9. In the non-oracle settings, Codex+DocPrompting did not improve over the base Codex; one explanation might be that the datasets are leaked into the training corpus of the Codex. For example, `CoNaLa` was extracted from StackOverflow, which is included in the large CommonCrawl corpus[8] that was used to train GPT-3, and possibly Codex. Therefore, Codex might have memorized the target code, and thus did not need the additional documentation. Although the data leakage issue might have happened in *code-davinci-001* as well, we suspect that this issue has worsened in the stronger *002* version. Regardless, we believe that the large capacity of Codex requires an equally strong retriever to improve over the base model. With an oracle retriever, DocPrompting yields significant improvement on both datasets. Thus, the non-oracle results could be further improved using a stronger non-oracle retriever.

## I    EXAMPLES

### I.1    TLDR

Examples on `tldr` are in Table 10 . In the top three cases, the baseline T5 could not generate the correct bash command while T5+DocPrompting retrieves the correct bash documentation and use the correct arguments. In the bottom two cases, T5 also predicts the bash command correctly. However, in the `puppet` case, it misses concrete usages of executing the code. In the `elixir` case, it halluciate the argument flags `--example` and `--arguments` which `elixir` does not support at all. On the other hand, T5+DocPrompting does ground its generation to the retrieved docs and therefore it generates more accurate code.

### I.2    CoNaLa

More examples on `CoNaLa` are shown in Table 11.

---

[8]`https://commoncrawl.org/the-data/`

Table 10: Examples of predictions from `tldr`, of the base T5 compared to T5+DocPrompting. In `tldr`, the bash commands in the test set are completely different from those in the training set.

| | |
|---|---|
| play a midi file in a specific key (0 = c major/a minor, 1 = f major/d minor, +1 = g major/e minor, etc.) | |
| Ground truth: | `timidity --force-keysig='-flats|+sharps' 'path/to/file.mid'` |
| T5: | `midiplay 'key'` |
| T5+DocPrompting: | `timidity --force-keysig 'f|f|f' 'path/to/file.midi'` |
| clean up all temporary tex files in the current directory | |
| Ground truth: | `latexmk -c` |
| T5: | `tex clean` |
| T5+DocPrompting: | `latexmk -c` |
| specify the hash algorithm for the index file (experimental) | |
| Ground truth: | `git show-index --object-format='sha1|sha256' 'path/to/file'` |
| T5: | `hash --experimental 'path/to/index.hashes'` |
| T5+DocPrompting: | `git show-index --object-format= 'path/to/hash-algorithm' 'path/to/experimental'` |
| execute puppet code | |
| Ground truth: | `puppet apply --execute 'code'` |
| T5: | `puppet` |
| T5+DocPrompting: | `puppet apply --execute 'path/to/file.puppet'` |
| evaluate elixir code by passing it as an argument | |
| Ground truth: | `elixir -e 'code'` |
| T5: | `elixir --example --arguments 'path/to/file.elixir'` |
| T5+DocPrompting: | `elixir -e 'path/to/file.elixir'` |

Table 11: Examples of predictions from `CoNaLa`, of the base CodeT5 compared to CodeT5+DocPrompting. Unseen functions are underscored.

| | |
|---|---|
| set the current working directory to `'c:\Users\uname\desktop\python'` | |
| Ground truth: | `os.chdir('c:\Users\uname\desktop\python')` |
| CodeT5: | `os.system('c:\Users\uname\desktop\python')` |
| CodeT5+DocPrompting: | `os.chdir('c:\Users\uname\desktop\python')` |
| convert dataframe 'df' to integer-type sparse object | |
| Ground truth: | `df.to_sparse(0)` |
| CodeT5: | `np.isinstance(df, np.integer)` |
| CodeT5+DocPrompting: | `df.to_sparse('i')` |

