# OpenReview forum: "DocPrompting: Generating Code by Retrieving the Docs"
_ICLR.cc/2023/Conference — ICLR 2023 notable top 25%_

### Official Review · Reviewer_k6bL · 2022-10-16

**Confidence:** 4
**Correctness:** 4
**Technical Novelty And Significance:** 3
**Empirical Novelty And Significance:** 4
**Recommendation:** 8

**Clarity, Quality, Novelty And Reproducibility:**

- Clarity: Most things are explained clearly either in the main paper or the appendix, except for the few points that are raised in the Weaknesses above.
- Quality: Well-designed and experiments and well-written work.
- Novelty: There is novelty in being the first paper to retrieve documentation for code generation and also in terms of the two newly curated datasets. However, in terms of technical novelty, already existing models and techniques are used for retrieval and generation.
- Reproducibility: Model training details are provided in the Appendix, which should be sufficient for reproducing these results, once the datasets are made available.


**Strength And Weaknesses:**

Strengths:
- The motivation and approach is well-grounded
- There are extensive results, spanning many different metrics. The Recall_unseen metric is particularly useful in demonstrating how much the prompting could help in a realistic scenario in which models are tested when new API changes are made on unseen functions.
- The analysis in Section 6 is very thorough, providing possible explanations for a number of different observations that are made.

Weaknesses:
- Adding retrieval will likely increase latency. The authors do not discuss/report how this is affected. As the size of the document pool increases, it seems that model latency will also increase.
- While the reason for splitting paragraphs into individual documents is explained in the Appendix, it seems odd to treat them as independent documents. For example, in Figure 2, the two bottom documents cannot standalone without the top one.
- It is not clear how k is selected for the number of documents to retrieve. Is it always 3?
- The annotation process for writing the test cases is not clear. In Appendix B, it is written that the two first authors did this. It is not clear whether each example was annotated by both of them and if so, what the agreement on that was.
- Missing reference: https://arxiv.org/pdf/1808.09588.pdf


**Summary Of The Paper:**

This paper presents the first approach for retrieving code documentation for code generation as a way of addressing the fact that APIs are constantly changing. They present a general strategy entailing a retriever (sparse-BM25 or dense-large pretrained model) which first retrieves a set of relevant documents from a pool of documents (which does not need to be fixed). Then, the retrieved documents and concatenated with the NL intent to form the prompt that is fed into a generation model for code generation. They evaluate a number of different generation models, including T5, CodeT5, GPT-Neo, and Codex. For evaluation, they curate two different datasets, by ensuring disjoint training/validation/test splits, spanning two different programming languages: bash commands (tldr) and Python (re-splitting CoNaLa). Their results show that in general, their DocPrompting approach tends to achieve superior performance over simply prompting with the NL intent alone.

Contributions:
- First paper to consider retrieving documentation for code generation
- Two curated datasets that can be useful for future work
- Extensive empirical experiments and ablation studies demonstrating the utility of their approach


**Summary Of The Review:**

While the technical novelty may be limited, there is novelty in terms of application and benchmark creation which will be useful for future work. The experiments and empirical results are thorough. Overall, I feel that this is a good paper contributing useful ideas and artifacts to the community.

---

> ### Author Response · Authors · 2022-11-10
> **Reply**
>
> Thank you for taking the time to review our paper and for your kind words!  We are happy to read that you enjoyed the well-grounded motivation and thorough experiments.
>
> We think that all your questions are addressable within this discussion period. Please see our response below. We will love to address additional questions during the discussion period if anything is unclear.
>
> > ### Does retrieval increase latency?  As the size of the document pool increases, it seems that model latency will also increase.
>
> Yes, retrieval increases latency somewhat, but using best practices for retrieval-based models the latency is not prohibitive. First, encoding the input for the retrieval step “costs” a *single forward pass* through the retriever’s encoder, which is significantly less expensive than generation (which requires multiple time steps of the decoder). All the documents in the retrieval pool can be encoded in advance, and finding the top-k results can be performed quickly using libraries such as [FAISS](https://github.com/facebookresearch/faiss) on the GPU or [ScaNN](https://github.com/google-research/google-research/tree/master/scann) on the CPU. The cost of this top-k search is sub-linear in the size of the document pool.
>
> The additional input to the *generator* results in an increased computational cost of encoding and a small increase in the computational cost of decoding, but latency increase can be limited by encoding the tokens of a given input in parallel.
>
> If this difference is crucial in practical settings, we can decrease the number of retrieved documents. To test this, we performed the experiments shown in Figure 3, which showed that retrieving as few as five docs may be sufficient in many cases.
>
> We have added this discussion in Appendix F.
>
>
> > ### it seems odd to treat individual paragraphs as independent documents.
> In tldr, we performed a *two-stage* retrieval where we first retrieve the entire bash manual and then retrieved the individual paragraphs of flags and arguments for the retrieved bash manual. That is, we only treated individual paragraphs as independent documents *after* deciding which bash command to use.  We have clarified this in Section 4.1 of the paper.
>
> In CoNaLa, we treat each function as an independent document, that can have multiple paragraphs.
>
> In our experiments, we found these approaches to perform better than other alternatives.
>
> > ### What is the number of retrieved docs ($k$)
>
> We used $k=10$ in all experiments as we described at the beginning of Section 5. This $k$ was tuned according to the validation set and to the available context window size of the models.  See also an analysis of the value of  $k$ in Figure 5 in Appendix F. We added a note in the caption of Table 1 and Table 3.
>
>
> > ### How many annotators wrote the unit tests?
>
> We followed the annotation process of the HumanEval dataset, and we had a single annotator for every example.
> Every annotator wrote multiple unit tests for their assigned example. “Agreement” is not relevant here, because there can be an infinite number of different possible unit tests, and we verified that the target code passes all our unit tests, so all of them are “correct” (differently from human annotations in other NLP tasks, where annotators can make mistakes, and the “correct” annotation might be ambiguous).
> We have clarified this in Appendix B of the revised paper.
>
> > ### Missing reference: “Mapping Language to Code in Programmatic Context”
>
> Thank you for pointing this out, we included this missing reference in our revised version.
>
> > ### in terms of technical novelty, already existing models and techniques are used for retrieval and generation.
>
> We acknowledge that the proposed method is relatively simple, leveraging widely tested techniques, but we also believe that methodological simplicity is often a strength rather than a weakness, as it allows for easier adoption. Our proposed approach is general and the retriever and the generator can be instantiated in various ways, which we show empirically (and as was acknowledged by the reviewers). We believe that this generality and simplicity make our approach *more* valuable to the readers, and will also make it useful for *future* retrievers and generators.

---

### Official Review · Reviewer_6cUe · 2022-10-16

**Confidence:** 4
**Correctness:** 4
**Technical Novelty And Significance:** 2
**Empirical Novelty And Significance:** 3
**Recommendation:** 8

**Clarity, Quality, Novelty And Reproducibility:**

## Novelty

### Overly-strong claims about existing models not being able to generalize *(addressed by new revision)*
The introduction makes very strong claims:
- "all existing code generation models either learn directly from input-output pairs provided as training data ... or learn the mapping between input and output implicitly from naturally occuring corpora"
- "all these works assume that all libraries and function calls were seen in the training data; and that at test time, the trained model will need to generate only seen libraries and function calls"
- "Thus, existing models inherently cannot generalize"

These claims seem much too strong. Although there has been a lot of work in the setting the authors discuss, there have also been a number of works that do not fit neatly into this categorization, and that allow models to generate code using new functions and libraries. For instance:

- [Hashimoto et al. (2018)](https://proceedings.neurips.cc/paper/2018/file/cd17d3ce3b64f227987cd92cd701cc58-Paper.pdf) and [Lu et al. (2017)](https://arxiv.org/pdf/1705.09042.pdf) learn to retrieve from a dataset of examples at inference time, which could be distinct from those used at training time. Some of the other works the authors cite also do something similar (e.g. Parvez et al. (2021) and Pasupat et al. (2021))
- [Shrivastava et al. (2021)](https://arxiv.org/abs/2206.12839) show that code generation can be improved by conditioning on context extracted from other relevant files, which may include new functions
-  [Wu et al. (2022)](https://arxiv.org/pdf/2203.08913.pdf) use an external memory to allow transformers to memorize new data at training time and recall it later, and show that this allows it to predict newly-defined functions at test time.

### Novelty of the approach and findings
I am not aware of prior work which specifically retrieves documentation to generate code from natural language, although there has been prior work on both *retrieving relevant documentation* (e.g. section 5.6 of [this survey](https://dl.acm.org/doi/pdf/10.1145/3212695)) and on *retrieving existing code to generate code* (e.g. [Hashimoto et al. (2018)](https://proceedings.neurips.cc/paper/2018/file/cd17d3ce3b64f227987cd92cd701cc58-Paper.pdf)).
As such, the empirical results provided by the authors do provide new evidence that documentation helps for code generation across a number of model architectures and tasks (as one might expect).
Additionally, the tldr task, the resplit of CoNaLa, and the documentation pool from DevDocs are new contributions of this work, and might be useful for evaluating improved documentation-retrieval models in the future.

Beyond this, the technique itself appears to be essentially the same as prior work: they use standard retrieval techniques based on existing algorithms and pre-trained models, and their generation models are also models that have been used for code or text generation before.

## Clarity
Overall, the paper is clearly written.

One thing I thought could be improved: Section 3.1 starts from the assumption that we are given triplets $(n, c, \mathcal{D}_n^*)$ where $n$ is a natural-language intent, $c$ is the target code, and $\mathcal{D}_n^*$ is the "ground truth docs" for $n$. I didn't think this was very well explained, since it's not obvious why $n$ would have ground truth documentation associated with it. From the rest of the paper and the appendix, it appears that $\mathcal{D}_n^*$ is actually selected based on a heuristic that uses $c$, not $n$. I think it would be better to explicitly describe this process in section 3.1, e.g. by stating that we are given a pair $(n, c)$ and then we infer a set of reference docs $\mathcal{D}_n^*$ from $c$. (I also wonder whether "ground truth docs" is the best terminology to use, since it's still being inferred heuristically and not manually observed or labeled.)  *(update: this has been addressed by the new revision.)*

Other minor suggestions:  *(update: addressed by the new revision.)*

- Section 4.2 states that ground-truth docs are constructed by "matching the function name with its documentation". However, there may be multiple function names in an example, which makes this statement ambiguous.
- I found page 7 hard to read, since it has a table at the top and a figure in the middle, with ordinary text interleaved around it.
- Figure 3 has overlapping text, and the axes are not labelled (I assume the x-axis is supposed to be $k$?)

## Quality
The dataset setup and experiments, which make up the bulk of the contribution, appear to be high quality overall. The  tldr dataset is an interesting source of documentation for bash code, and re-splitting the CoNaLa dataset based on use of new functions is a good idea for assessing generalization ability.

Experimentally, the authors do a good job of evaluating DocPrompting across a variety of metrics and for a variety of base models. They also conduct a series of interesting ablations and extensions, including using oracle access to the appropriate documentation, comparing to example-based prompting, and investigating different retrievers.

One potential concern is that the strongest results on CoNaLa are given by CodeT5, but CodeT5 was pretrained on a dataset of code that might include some of the "held-out" Python functions described in 4.2. This might be a source of knowledge leakage, since CodeT5-based retrievers might just pick documentation that includes functions that accomplish the task by simply memorizing the functions. It would be helpful to analyze whether this leakage might have occurred, and if so, how much of an effect it might have on the results in Tables 3 and 4. (Am I right in assuming all of the results in Table 3 used a CodeT5-based retriever?)  *(update: Authors have added some discussion of this in the appendix.)*

Additionally, I wasn't convinced by the argument in section 6.1. If I understand correctly, this section compares n-gram overlap between the natural language $n$ and code $c$, and also n-gram overlap between the "ground-truth" docs $\mathcal{D}_n^*$ (along with $n$) and the code $c$. But, from the appendix, $\mathcal{D}_n^*$ is selected exactly by finding the documentation with the highest string overlap with $c$! So it seems obvious that $\mathcal{D}_n^*$ and $c$ would have high n-gram overlap, regardless of how much documentation does or doesn't "ease the mapping between NL and code". (A better experiment might be to look at n-gram overlap between $c$ and the *retrieved* examples from the model, using only information from $n$ instead of the oracle. It would also be interesting to look at overlap between the NL description and the documentation to make sure the gap is actually being "bridged" on both sides.)  *(update: The authors have taken my suggestion here and presented results using the retrieved documents.)*

## Reproducibility
The technique is fairly straightforward, and the authors release their new benchmarks and datasets, so it seems likely this work will be reproducible.

**Strength And Weaknesses:**

Strengths:

- Using documentation as an extra input to generating code is a good idea for ensuring models generalize to new types of functions, and this paper confirms that this generalization is indeed enabled by this approach.
- The authors compare their approach with a variety of alternatives and find a consistent improvement across experiment settings.
- The new datasets/benchmarks are interesting and seem like a reasonable way to evaluate generalization to unseen types of function.

Weaknesses:

- ~~The novelty is primarily empirical in nature, and~~ the technical approach is essentially an application of known techniques. *(update: After reflection and discussion with the authors, I admit that there is novelty in the problem formulation as well, which goes beyond the setting considered by previous work.)*
- ~~The introduction makes bold claims about "all existing code generation models" which I do not think are justified.~~ *(update: The unsupported claims have been qualified in the new revision.)*
- ~~The notion of "ground truth docs" is a bit confusing.~~ *(update: The paper has been reworded to make this more clear.)*
- It's possible there is some function-knowledge leakage due to the use of pretrained code models for retrieval (perhaps the authors can clarify this) *(update: I am still curious whether this influences the results, but it is a bit orthogonal to the main point of the paper, since it would also affect baselines. The authors have added a brief discussion of this potential leakage to the appendix.)*


**Summary Of The Paper:**

This paper describes an approach which uses documentation to generate code conditioned on a natural-language (NL) intent. The approach is essentially a standard retrieval-conditioned generation strategy, but where the set of documents being retrieved are constructed based on documentation.

The paper describes a set of practical instantiations of this technique, exploring a variety of retrieval models (sparse heuristic-based retrievers, and "dense" learned retrieval models trained to recover "correct" docs), and a few variants of generator architectures based on transformers, with either fine-tuning or few-shot prompting. The paper also introduces two datasets: a new benchmark for NL->bash prediction based on the `tldr` docs, and a variant of the CoNaLa benchmark with a different test split combined with a new reference corpus of documentation based on `devdocs.io`.

The authors show that retrieving documentation improves performance across all model types, and also study the effects of retrieving documentation vs code, providing some ground-truth knowledge, and using different retriever strategies.

**Summary Of The Review:**

*Original review:* Using documentation as additional context for generating code is a sensible idea, and it is good to have a thorough empirical confirmation that this does in fact yield higher accuracy in the NL -> code setting. Thus, although the novelty of the technique itself is limited, and the findings aren't particularly surprising, I tend to lean toward acceptance (assuming the issues I mention regarding strength of claims, clarity, dataset leakage, and n-gram overlap can be resolved).

*Updated review:* The authors have resolved my concerns, and upon reflection I do think that the problem formulation is interesting regardless of the novelty in the technique itself. I have increased my score from 6 to 8.

---

> ### Author Response · Authors · 2022-11-10
> **Reply 1/2**
>
> Thank you for taking the time to review our paper, for your constructive feedback, and for your kind words! We are happy to read the feedback that *using documentation is a good idea for ensuring models generalize to new types of functions*.
>
> We think that all your questions are addressable within this discussion period. Please see our response below. We will be happy to address additional questions during the discussion period if anything is unclear.
>
>
> > ### The technical approach is essentially an application of known techniques.
>
> We acknowledge that the proposed method is relatively simple, leveraging widely tested techniques, but we also believe that methodological simplicity is often a strength rather than a weakness, as it allows for easier adoption. Our proposed approach is general and the retriever and the generator can be instantiated in various ways, which we show empirically (and as was acknowledged by the reviewers). We believe that this generality and simplicity make our approach *more* valuable to the readers, and will also make it useful for *future* retrievers and generators.
>
>
> > ### It's possible there is some function-knowledge leakage due to the use of pretrained code models for retrieval (perhaps the authors can clarify this)
> > ### One potential concern is that the strongest results on CoNaLa are given by CodeT5 … since CodeT5-based retrievers might just pick documentation that includes functions that accomplish the task by simply memorizing the functions.
>
> It is possible that some functions were seen during the pretraining of CodeT5, however, the memorization issue is likely negligible:
> Even if the “NL intents” or the documentation were seen during pretraining - **they were never seen together**, and thus the *matching* between NL$\leftrightarrow$doc was certainly not leaked. Since this matching was not leaked, it could not have helped the retriever, which predicts this matching.
>
>
> Further evidence is shown by the analysis in Table 4:
> When using off-the-shelf CodeT5 retrievers on CoNaLa (Table 4), recall@1 *for unseen functions* is as low as 4.60, compared to 16.54 after our contrastive training of the CodeT5 retriever.
>
> > ### Some claims are overly-strong - some prior work could generalize beyond the training data by retrieving examples (Hashimoto et al. 2018, Lu et al. 2017) and code snippets from other relevant files (Shrivastava et al. 2021, Wu et al. 2022)
>
> We agree, we have softened the claims to “all existing **non-retrieving** code generation models... assume that all libraries and function calls were seen in the training data; and that at test time, the trained model will need to generate only seen libraries and function calls” in our revised version. We also included a discussion that explains the differences from all these missing references in the Related Work section.
>
> However, we do believe that when predicting new libraries and functions that are *external* to the user’s project, retrieving docs is probably the most practical approach. Further, we believe that documentation for the new libraries is much more likely to be available than concrete (natural language intents, code snippet) pairs or individual code snippets that use these libraries already.
>
>
> > ### Are there “ground truth docs”? Is “ground truth docs” the best terminology?
>
> We agree that it is a bit confusing. The “ground truth docs” are the documentation of functions that are used in the ground truth code. We referred to these docs as “ground truth docs” because they were obtained using the target code $c$, rather than the input NL intent $n$, and eventually, these are the docs that we train the model to retrieve. In our revised version, we have renamed these docs to **oracle docs** to avoid confusion.
>
> We used heuristics to infer these oracle docs instead of annotating them manually because in our manual examination, these heuristics work well except for a few corner cases which we fixed as well.
>
> We note that inferring these “oracle docs” during the data creation process was relatively easy, while retrieving them at test time is hard. The reason is that we used the target code $c$ to infer the oracle docs when creating the dataset, but $c$ is not available at test time.
>
>
>
> > ### What are the “ground truth docs” when there are multiple function names in an example?
>
> The ground truth (oracle) docs of a given code snippet are the *union* of the docs of all functions and arguments that are used in the code snippet. We clarified this in our revised version.

---

> > ### Author Response · Authors · 2022-11-10
> > **Reply 2/2**
> >
> > > ### ​​A better experiment might be to look at the n-gram overlap between $c$ and the *retrieved* examples from the model, using only information from $n$ instead of the oracle
> > > ### It would also be interesting to look at the overlap between the NL description and the documentation to make sure the gap is actually being "bridged" on both sides.
> >
> > Thanks for the suggestion. We agree that measuring the n-gram matching between the code and the *retrieved* docs is more meaningful because no oracle information is considered. Here we show the n-gram matching of (NL$\leftrightarrow$Code), (NL+*retrieved* docs$\leftrightarrow$Code) and (NL$\leftrightarrow$retrieved docs)
> >
> > | tldr                        | 1  | 2 | 3 |
> > |-----------------------------|----|---|---|
> > | NL$\leftrightarrow$code                  | 12 | 0 | 0 |
> > | (NL + retrieved docs)$\leftrightarrow$code | 24 | 2 | 0 |
> > | NL$\leftrightarrow$retrieved docs |  39 | 8 | 3 |
> >
> >
> > | CoNaLa                      | 1  | 2  | 3  | 4  | 5 |
> > |-----------------------------|----|----|----|----|---|
> > | NL$\leftrightarrow$code                  | 30 | 14 | 11 | 9  | 7 |
> > | (NL + retrieved docs)$\leftrightarrow$code | 91 | 52 | 28 | 16 | 11 |
> > | NL$\leftrightarrow$retrieved docs |  72 | 14 | 3 | 1| 1 |
> >
> > We can see that having *retrieved* docs also significantly increases the n-gram overlap compared to NL$\leftrightarrow$code (the n-gram matching is sometimes even higher than when using the oracle docs, because there are 10 retrieved docs and only 2-3 oracle docs). We have updated this analysis in Figure 4 in the paper, to show the retrieved docs rather than the oracle docs.
> >
> > Since we used BM25 to retrieve docs in tldr, the NL$\leftrightarrow$retrieved docs overlap is high by construction; in CoNaLa, the NL$\leftrightarrow$retrieved docs unigram overlap is high as well, but since we used a *dense* retriever, the general n-gram overlap does not have to be high for our approach to work well.
> >
> >
> >
> > > ### Am I right in assuming all of the results in Table 3 used a CodeT5-based retriever?
> >
> > Yes, we have clarified this in the caption of Table 3 in our revised version.
> >
> > > ### Other minor suggestions ...
> >
> > Thank you for all these presentation comments. We have fixed all of them in our revised version.

---

> > > ### Comment · Reviewer_6cUe · 2022-11-13
> > > **Response to author reply and new revision**
> > >
> > >
> > > > > The technical approach is essentially an application of known techniques.
> > > >
> > > > We believe that this generality and simplicity make our approach more valuable to the readers, and will also make it useful for future retrievers and generators.
> > >
> > > My point here was not about generality or simplicity (which I agree are useful) but instead about novelty of the technique relative to existing work. (In terms of generality, previous retrieval techniques could be considered more general and simpler than the proposed approach, because they include it as a special case.) It also seems fairly easy for someone else to independently think of applying these existing techniques to documentation. It seems to me that the most significant contribution here for the community is the empirical evidence that such a technique works well in practice and helps with generalization to new functions.
> > >
> > > > > It's possible there is some function-knowledge leakage
> > > >
> > > > ... the memorization issue is likely negligible: Even if the “NL intents” or the documentation were seen during pretraining - they were never seen together, and thus the matching between NL <-> doc was certainly not leaked. Since this matching was not leaked, it could not have helped the retriever, which predicts this matching.
> > >
> > > I don't think this is necessarily true. CodeT5 likely did see combinations of *NL intent* and *code* (e.g. from comments), and documentation contains code. So, it's fairly plausible to me that off-the-shelf CodeT5 would be able to associate "NL intent X" with "documentation that contains code C that solves intent X" without training on matched pairs, by virtue of associating "NL intent X" with "code C that solves intent X" directly during pretraining. (For example, if CodeT5 already knows to associate NL intents like "plot this data" with code like "np.plot", it would be able to correctly retrieve documentation by memorizing and looking for "np.plot" in the code snippets, instead of by looking for NL descriptions that match "plot this data". But this strategy would not work for a new function that CodeT5 hasn't already learned to look for.)
> > >
> > > Thanks for pointing out the results in Table 4. Although this provides some evidence that leakage plays a relatively small role (if any) in the accuracy of the learned CodeT5-based retriever, it's still plausible to me that the pretraining mechanism could expose some latent memorized knowledge that the model already has, so I don't think we can say for certain. (The ideal experiment would be to evaluate on functions that were not seen even in the pretraining of CodeT5, but that admittedly seems quite tricky. A slightly easier, but perhaps still impractical, experiment would be to see if CodeT5 can retrieve the oracle documentation based only on the code snippets in that documentation, without looking at the accompanying documentation or comments.)
> > >
> > > Given how much the experimental setup section is focused on preventing leakage, I still think it would be worthwhile to at least mention this possibility somewhere in the paper.
> > >
> > > > > Some claims are overly-strong
> > >
> > > Thanks for qualifying the claims in the introduction, although they now seem a bit tautological, since it seems to now just say that non-retrieving models weren't trained using retrieval (?). Perhaps something like "Many existing code generation models" would better get the point across without claiming to be exhaustive.
> > >
> > > Thanks also for including citations of the related work. I agree that retrieving documentation is likely to be a more practical strategy than some of the other approaches considered previously.
> > >
> > > > we have renamed these docs to oracle docs to avoid confusion.
> > >
> > > Thanks, referring to them as oracle docs is much clearer.
> > >
> > > > n-gram overlap between  and the retrieved examples
> > >
> > > Thanks for including the new $n$-gram results in your response. It would be good to also include these tables in the paper as well, perhaps in the appendix (in particular, although figure 4 was updated, I don't think the "NL <-> retrieved docs" row is currently included).

---

> > > > ### Author Response · Authors · 2022-11-16
> > > > **Response**
> > > >
> > > > Thank you for reading our response and for your devotion. Please see our reply below.
> > > >
> > > > > ### It also seems fairly easy for someone else to independently think of applying these existing techniques to documentation. It seems to me that the most significant contribution here for the community is the empirical evidence that such a technique works well in practice and helps with generalization to new functions.
> > > >
> > > > It might be fairly easy for someone else to independently think of using documentation, but in practice, no other paper has used documentation this way nor recognized the problem of “generalization to future libraries”, despite the popularity of this field in recent years.
> > > >
> > > > Thus, we believe that our contribution is not only the empirical evidence but also the recognition of the *problem* and the insight of using documentation as a (possible) *solution*.
> > > >
> > > >
> > > > > ### Discussion regarding leakage
> > > >
> > > > Thank you for this suggestion and the example, we reflected this discussion in Appendix B due to space constraints.
> > > >
> > > > > ### Thanks for qualifying the claims in the introduction, although they now seem a bit tautological, since it seems to now just say that non-retrieving models weren't trained using retrieval (?). Perhaps something like "Many existing code generation models" would better get the point across without claiming to be exhaustive.
> > > >
> > > > Thank you for this suggestion, we have rephrased that part in Section 1.
> > > >
> > > > Notice, however, that the previous phrasing did *not* say that “non-retrieving models weren’t trained using retrieval”. Instead, we highlight that non-retrieving models suffer from the “generalization to future libraries” problem, and this insight is *not* trivial to the broader audience.
> > > >
> > > > > ### Thanks for including the new $n$-gram results in your response. It would be good to also include these tables in the paper as well, perhaps in the appendix (in particular, although figure 4 was updated, I don't think the "NL <-> retrieved docs" row is currently included).
> > > >
> > > > Thank you for noticing, we included these numbers in Table 8, and added the discussion in Appendix F.

---

> > > > > ### Comment · Reviewer_6cUe · 2022-11-17
> > > > > **Updated review**
> > > > >
> > > > > True, that is a good point that there is value in a novel formulation of the problem, independent of the novelty in the technique itself; my earlier statement about it being a mostly empirical contribution was a bit too reductive. Thanks also for adding the additional discussion of leakage and of the n-gram results.
> > > > >
> > > > > I have updated my review and increased the score from 6 to 8.

---

> > > > > > ### Author Response · Authors · 2022-11-17
> > > > > > **Thank you**
> > > > > >
> > > > > > Thank you for your valuable suggestions in this enjoyable discussion and for raising your score!

---

### Official Review · Reviewer_145E · 2022-10-24

**Confidence:** 3
**Correctness:** 3
**Technical Novelty And Significance:** 3
**Empirical Novelty And Significance:** 3
**Recommendation:** 8

**Clarity, Quality, Novelty And Reproducibility:**

The presentation is of high quality and the paper is well written. The idea of retrieving documentation for prompting seems interesting and novel to me.

**Strength And Weaknesses:**

Strengths:
* The paper has a clear motivation and is easy to follow.
* The idea of document prompting is tested across a range of models and retrievers with promising results. The authors also compare DocPrompting to a baseline with in-context examples (ExPrompting) instead of in-context documentation (DocPrompting).
* The n-gram analysis provides a simple account of how DocPrompting may bring benefits.

Weaknesses:
* I am unsure how fair the evaluation setting with unseen functions is for the baseline models, especially since the ExPrompting baseline cannot retrieve examples of unseen functions from the training set.

**Summary Of The Paper:**

The authors propose DocPrompting for the problem of generating code from natural language instructions. DocPrompting supplies the model with in-context documentation and uses retrieval to find relevant documentation for a given input. The authors experiment with using BM25 and training dense encoders for retrieving documentation, and find consistent improvements with DocPrompting. The authors focus on evaluation settings where functions in the test set are not seen during training, and also introduce the new code generation dataset “tldr”.

**Summary Of The Review:**

This paper proposes the simple and intuitive idea of incorporating documentation into code generation tasks and finds that it brings empirical benefits, especially when generalizing to unseen functions. The paper is well executed and will likely lead to future work.

---

> ### Author Response · Authors · 2022-11-10
> **Reply**
>
> Thank you for taking the time to review our paper and for your kind words!  We are happy for the encouraging words about the clear motivation for the paper.
>
> Please see our response below. We would be happy to address additional questions during the discussion period if anything is unclear.
>
> > ### the ExPrompting baseline cannot retrieve examples of unseen functions from the training set
>
> When new libraries are released they often come with documentation, and thus we can assume that documentation for new libraries is much more likely to be available than concrete (natural language command, code snippet) pairs, or simply code snippets that use these libraries already. Our experiment with ExPrompting mainly simulates this scenario.

---

### Official Review · Reviewer_xC2U · 2022-10-24

**Confidence:** 4
**Clarity, Quality, Novelty And Reproducibility:** Please see the Strength And Weaknesses
**Correctness:** 4
**Technical Novelty And Significance:** 3
**Empirical Novelty And Significance:** 4
**Recommendation:** 6

**Strength And Weaknesses:**

Strengths:
1. The idea of leveraging the newly added manuals and documentation to generate code containing unseen and unused functions and libraries is interesting and novel. The work demonstrates an exciting endeavor direction to generate high-quality code.
2. The main results from Tables 1 & 3 demonstrate that the improvement of the proposed DocPrompting is significant across various pre-trained models. Conducting experiments on more than one benchmark and thorough ablations make the conclusion of this work convincing.
3. The presentation of this work is excellent. The rich use of diagrams and tables makes this paper easy to understand.

Weaknesses:
1. Although the usage of documentation for code generation is novel, it is somewhat simple from a technical point of view.
2. I examined the supplementary material provided by the authors and only found the data files. It would be better if there were codes to prove its reproducibility.

**Summary Of The Paper:**

This paper observes that existing models can only generate seen libraries and function calls at test time. However, new functions and libraries are introduced all the time, and even a seen function call can have arguments that were not used in the training data. To address such unseen functions in code generation, the authors propose a simple and effective approach for code generation by retrieving the relevant documentation, dubbed DocPrompting. The extensive experiments demonstrate that DocPrompting consistently improves NL->code models in two tasks, in two PLs, and across multiple strong base models. DocPrompting improves strong base models such as CodeT5 by 2.85% in pass@1 (52% relative gain) in execution-based evaluation on the popular Python CoNaLa benchmark; on a new Bash dataset tldr, DocPrompting improves CodeT5 and GPT-Neo-1.3B by up to 6.9% exact match, and Codex by 6.78 charBLEU score.

**Summary Of The Review:**

In my opinion, the proposed DocPrompting method is interesting and novel, and the experimental results successfully demonstrated the effectiveness of the proposed method. Considering the technical simplicity and reproducibility, I give a weak accept.

---

> ### Author Response · Authors · 2022-11-10
> **Reply**
>
> Thank you for taking the time to review our paper and for your kind words! We are very happy to hear the encouraging words about the work being interesting, novel, and exciting.
>
> Please see our response below. We would be very happy to address additional questions during the discussion period if anything is unclear.
>
> > ### The usage of documentation for code generation is novel, but it is simple from a technical point of view.
>
> We acknowledge that the proposed method is relatively simple, leveraging widely tested techniques, but we also believe that methodological simplicity is often a strength rather than a weakness, as it allows for easier adoption. Our proposed approach is general and the retriever and the generator can be instantiated in various ways, which we show empirically (and as was acknowledged by the reviewers). We believe that this generality and simplicity make our approach *more* valuable to the readers, and will also make it useful for *future* retrievers and generators.
>
>
> > ### The authors included the datasets, but did not include their code yet
>
> We will release all code, data, and trained models. Our code is based on Huggingface transformers, and we will make all datasets and trained models available on the Huggingface hub.
>
>
> **Please let us know if you have any additional questions or concerns.**

---

### Decision · Program_Chairs · 2023-01-20

**Decision:**

Accept: notable-top-25%

**Justification For Why Not Higher Score:**

The scope is limited to codegen tasks.

**Justification For Why Not Lower Score:**

This is a valuable addition to the ICLR line-up, and the method is simple and applicable enough that it would be of interest to enough people.

**Metareview: Summary, Strengths And Weaknesses:**

The paper presents a method to generate code from natural language prompts by retrieving snippets of documentation that is relevant to the prompts and augmenting the codegen model prompt with those snippets. It is evaluated extensively on several existing benchmark, also introduces a new benchmark (for bash), and is compared to adequate baselines (ROBERTA, CodeT5) that it can augment itself (adding retrieval). It improves performance across the board. Possible improvements to the submission include a more thorough data leakage / retrieval advantage analysis in the results (even though no reviewer expects it would change the experimental conclusions), and more details for reproducibility. Overall, it is a simple method with a direct, visible effect on tasks of the codegen community.

**Note From Pc:**

if the above contains the word "oral" or "spotlight" please see: "oral" presentation means -> notable-top-5% and "spotlight" means -> notable-top-25%. As stated in our emails, we are disassociating presentation type from AC recommendations

**Summary Of Ac-Reviewer Meeting:**

N/A